# Learning spatiotemporal piecewise-geodesic trajectories from longitudinal manifold-valued data

**Juliette Chevallier**
CMAP, École polytechnique
juliette.chevallier@polytechnique.edu

**Pr Stéphane Oudard**
Oncology Department
USPC, AP-HP, HEGP

**Stéphanie Allassonnière**
CRC, Université Paris Descartes
stephanie.allassonniere@parisdescartes.fr

## Abstract

We introduce a hierarchical model which allows to estimate a group-average *piecewise-geodesic* trajectory in the Riemannian space of measurements and individual variability. This model falls into the well defined mixed-effect models. The subject-specific trajectories are defined through spatial and temporal transformations of the group-average piecewise-geodesic path, component by component. Thus we can apply our model to a wide variety of situations. Due to the non-linearity of the model, we use the Stochastic Approximation Expectation-Maximization algorithm to estimate the model parameters. Experiments on synthetic data validate this choice. The model is then applied to the metastatic renal cancer chemotherapy monitoring: we run estimations on RECIST scores of treated patients and estimate the time they escape from the treatment. Experiments highlight the role of the different parameters on the response to treatment.

## 1 Introduction

During the past few years, the way we treat renal metastatic cancer was profoundly changed: a new class of anti-angiogenic therapies targeting the tumor vessels instead of the tumor cells has emerged and drastically improved survival by a factor of three (Escudier et al., 2016). These new drugs, however, do not cure the cancer, and only succeed in delaying the tumor growth, requiring the use of successive therapies which must be continued or interrupted at the appropriate moment according to the patient's response. The new medicine process has also created a new scientific challenge: how to choose the more efficient drug therapy. This means that one has to properly understand how the patient reacts to the possible treatments. Actually, there are several strategies and taking the right decision is a contested issue (Rothermundt et al., 2015, 2017).

To achieve that goal, physicians took an interest in mathematical modeling. Mathematics has already demonstrated its efficiency and played a role in the change of stop-criteria for a given treatment (Burotto et al., 2014). However, to the best of our knowledge, there only exists one model which was designed by medical practitioners. Although, very basic mathematically, it seems to show that this point of view may produce interesting results. Introduced by Stein et al. in 2008, the model performs a non-linear regression using the least squares method to fit an increasing or/and decreasing exponential curve. This model is still used but suffers limitations. First, as the profile are fitted individual-by-individual independently, the model cannot explain a global dynamic. Then, the choice of exponential growth avoids the emergence of plateau effects which are often observed in practice. This opens the way to new models which would explain both a population and each individual with other constraints on the shape of the response.

Learning models of disease progression from such databases raises great methodological challenges. We propose here a very generic model which can be adapted to a large number of situations. For a given population, our model amounts to estimating an average trajectory in the set of measurements and individual variability. Then we can define continuous subject-specific trajectories in view of the population progression. Trajectories need to be registered in space and time, to allow anatomical variability (as different tumor sizes), different paces of progression and sensitivity to treatments. The framework of mixed-effects models is well suited to deal with this hierarchical problem. Mixed-effects models for longitudinal measurements were introduced in the seminal paper of Laird and Ware (1982) and have been widely developed since then. The recent generic approach of Schiratti et al. (2015) to align patients is even more suitable. First, anatomical data are naturally modeled as points on Riemannian manifolds while the usual mixed-effects models are defined for Euclidean data. Secondly, the model was built with the aim of granting individual temporal and spatial variability through individual variations of a common time-line grant and parallel shifting of the average trajectory.

However, Schiratti et al. (2015) have made a strong hypothesis to build their model as they consider that the mean evolution is a geodesic. This would mean in our targeted situation that the cancer would either go on evolving or is always sensitive to the treatment. Unfortunately, the anti-angiogenic treatments may be inefficient, efficient or temporarily efficient, leading to a re-progression of the metastasis. Therefore, we want to relax this assumption on the model.

In this paper, we propose a generic statistical framework for the definition and estimation of spatiotemporal piecewise-geodesic trajectories from longitudinal manifold-valued data. Riemannian geometry allows us to derive a method that makes few assumptions about the data and applications dealt with. We first introduce our model in its most generic formulation and then make it explicit for RECIST (Therasse et al., 2000) score monitoring, *i.e.* for one-dimension manifolds. Experimental results on those scores are given in section 4.2. The introduction of a more general model is a deliberate choice as we are expecting to apply our model to the corresponding medical images. Because of the non-linearity of the model, we have to use a stochastic version of the Expectation-Maximization algorithm (Dempster et al., 1977), namely the MCMC-SAEM algorithm, for which theoretical results regarding the convergence have been proved in Delyon et al. (1999) and Allassonnière et al. (2010) and numerical efficiency has been demonstrated for these types of models (Schiratti et al. (2015), MONOLIX – MOdèles NOn LInéaires à effets miXtes).

## 2 Mixed-effects model for piecewise-geodesically distributed data

We consider a longitudinal dataset obtained by repeating measurements of $n \in \mathbb{N}^*$ individuals, where each individual $i \in [\![1, n]\!]$ is observed $k_i \in \mathbb{N}^*$ times, at the time points $t_i = (t_{i,j})_{1 \leqslant j \leqslant k_i}$ and where $y_i = (y_{i,j})_{1 \leqslant j \leqslant k_i}$ denotes the sequence of observations for this individual. We also denote $k = \sum_{i=1}^{n} k_i$ the total numbers of observations. We assume that each observation $y_{i,j}$ is a point on a $d$-dimensional geodesically complete Riemannian manifold $(M, g)$, so that $y = (y_{i,j})_{1 \leqslant i \leqslant n, \, 1 \leqslant j \leqslant k_i} \in M^k$.

We generalize the idea of Schiratti et al. (2015) and build our model in a hierarchical way. We see our data points as samples along trajectories and suppose that each individual trajectory derives from a group-average scenario through spatiotemporal transformations. Key to our model is that the group-average trajectory in no longer assumed to be geodesic but piecewise-geodesic.

### 2.1 Generic piecewise-geodesic curves model

Let $m \in \mathbb{N}^*$ and $t_R = \left(-\infty < t_R^1 < \ldots < t_R^{m-1} < +\infty\right)$ a subdivision of $\mathbb{R}$, called the *breaking-up times* sequence. Let $M_0$ a $d$-dimensional geodesically complete manifold and $\left(\bar{\gamma}_0^\ell\right)_{1 \leqslant \ell \leqslant m}$ a family of geodesics on $M_0$. To completely define our average trajectory, we introduce $m$ isometries $\phi_0^\ell \colon M_0 \to M_0^\ell := \phi_0^\ell(M_0)$. This defines $m$ new geodesics – on the corresponding space $M_0^\ell$ – by setting down $\gamma_0^\ell = \phi_0^\ell \circ \bar{\gamma}_0^{\,\ell}$. The isometric nature of the mapping $\phi_0^\ell$ ensures that the manifolds $M_0^\ell$ remain Riemannian and that the curves $\gamma_0^\ell$ remain geodesic. In particular, each $\gamma_0^\ell$ remains parametrizable (Gallot et al., 2004). We define the average trajectory by

$$\forall t \in \mathbb{R}, \quad \gamma_0(t) = \gamma_0^1(t) \, \mathbb{1}_{]-\infty, t_R^1]}(t) + \sum_{\ell=2}^{m-1} \gamma_0^\ell(t) \, \mathbb{1}_{]t_R^{\ell-1}, t_R^\ell]}(t) + \gamma_0^m(t) \, \mathbb{1}_{]t_R^{m-1}, +\infty[}(t).$$

In this framework, $M_0$ may be understood as a *manifold-template* of the geodesic components of the curve $\gamma_0$.

Because of the piecewise nature of our average-trajectory $\gamma_0$, constraints have to be formulated on each interval of the subdivision $t_R$. Following the formulation of the *local existence and uniqueness theorem* (Gallot et al., 2004), constraints on geodesics are generally formulated by forcing a value and a tangent vector at a given time-point. However, such an approach cannot ensure the curve $\gamma_0$ to be at least continuous. That is why we re-formulate these constraints in our model as boundary conditions. Let a sequence $\bar{A} = (\bar{A}^0, \ldots, \bar{A}^m) \in (M_0)^{m+1}$, an initial time $t_0 \in \mathbb{R}$ and a final time $t_1 \in \mathbb{R}$. We impose[1] that for all $\ell \in [\![1, m-1]\!]$, $\bar{\gamma}_0^1(t_0) = \bar{A}^0$, $\bar{\gamma}_0^\ell(t_R^\ell) = \bar{A}^\ell$, $\bar{\gamma}_0^{\ell+1}(t_R^\ell) = \bar{A}^\ell$ and $\bar{\gamma}_0^m(t_1) = \bar{A}^m$. Notably, the $2m$ constraints are defined step by step. In one dimension (cf section 2.2), the geodesics could be written explicitly and such constraints do not complicate the model so much. In higher dimension, we have to use shooting or matching methods to enforce this constraint.

In practice, the choice of the isometries $\phi_0^\ell$ and the geodesics $\bar{\gamma}_0^\ell$ have to be done with the aim to be "as regular as possible" (at least continuous as said above) at the rupture points $t_R^\ell$. In one dimension for instance, we build trajectories that are continuous, not differentiable but with a very similar slope on each side of the breaking-points.

We want the individual trajectories to represent a wide variety of behaviors and to derive from the group average path by spatiotemporal transformations. To do that, we define for each component $\ell$ of the piecewise-geodesic curve $\gamma_0$ a couple of transformations $(\phi_i^\ell, \psi_i^\ell)$. These transformations, namely the *diffeomorphic component deformations* and the *time component reparametrizations*, characterize respectively the spatial and the temporal variability of propagation among the population. Thus, individual trajectories may write in the form of

$$\forall t \in \mathbb{R}, \quad \gamma_i(t) = \gamma_i^1(t) \, \mathbb{1}_{]-\infty, t_{R,i}^1]}(t) + \sum_{\ell=2}^{m-1} \gamma_i^\ell(t) \, \mathbb{1}_{]t_{R,i}^{\ell-1}, t_{R,i}^\ell]}(t) + \gamma_i^m(t) \, \mathbb{1}_{]t_{R,i}^{m-1}, +\infty[}(t) \quad (\star)$$

where the functions $\gamma_i^\ell$ are obtained from $\gamma_0^\ell$ through the applications of the two transformations $\phi_i^\ell$ and $\psi_i^\ell$ described below. Note that, in particular, each individual possesses his own sequence of rupture times $t_{R,i} = (t_{R,i}^\ell)_{1 \leqslant \ell < m}$. Moreover, we require the fewest constraints possible in the construction: at least continuity and control of the slopes at these breaking-up points.

For compactness, we will now abusively denote $t_R^0$ for $t_0$ and $t_R^m$ for $t_1$.

To allow different paces in the progression and different breaking-up times for each individual, we introduce some temporal transformations $\psi_i^\ell$, called *time-warp*, that are defined for the subject $i \in [\![1, n]\!]$ and for the geodesic component $\ell \in [\![1, m]\!]$ by $\psi_i^\ell(t) = \alpha_i^\ell(t - t_R^{\ell-1} - \tau_i^\ell) + t_R^{\ell-1}$. The parameters $\tau_i^\ell$ correspond to the time-shift between the mean and the individual progression onset and the $\alpha_i^\ell$ are the acceleration factors that describe the pace of individuals, being faster or slower than the average. To ensure good adjunction at the rupture points, we demand the individual breaking-up times $t_{R,i}^\ell$ and the time-warps to satisfy $\psi_i^\ell(t_{R,i}^\ell) = t_R^\ell$ and $\psi_i^\ell(t_{R,i}^{\ell-1}) = t_R^{\ell-1}$. Hence the subdivision $t_{R,i}$ is constrained by the time reparametrizations, which are also constrained. Only the acceleration factors $\alpha_i^\ell$ and the first time shift $\tau_i^1$ are free: for all $\ell \in [\![1, m]\!]$, the constraints rewrite step by step as $t_{R,i}^\ell = t_R^{\ell-1} + \tau_i^\ell + \frac{t_R^\ell - t_R^{\ell-1}}{\alpha_i^\ell}$ and $\tau_i^\ell = t_{R,i}^{\ell-1} - t_R^{\ell-1}$.

Concerning the space variability, we introduce $m$ diffeomorphic deformations $\phi_i^\ell$ which enable the different components of the individual trajectories to vary more irrespectively of each other. We just enforce the adjunction to be at least continuous and therefore the diffeomorphisms $\phi_i^\ell$ have to satisfy $\phi_i^\ell \circ \gamma_0^\ell(t_R^\ell) = \phi_i^{\ell+1} \circ \gamma_0^{\ell+1}(t_R^\ell)$. Note that the mappings $\phi_i^\ell$ do not need to be isometric anymore, as the individual trajectories are no longer required to be geodesic.

Finally, for all $i \in [\![1, n]\!]$ and $\ell \in [\![1, m]\!]$, we set $\gamma_i^\ell = \phi_i^\ell \circ \gamma_0^\ell \circ \psi_i^\ell$ and define $\gamma_i$ as in $(\star)$. The observations $y_i = (y_{i,j})$ are assumed to be distributed along the curve $\gamma_i$ and perturbed by an additive Gaussian noise $\varepsilon_i \sim \mathcal{N}(0, \sigma^2 I_{k_i})$ :

$$\forall (i,j) \in [\![1, n]\!] \times [\![1, k_i]\!], \quad y_{i,j} = \gamma_i(t_{i,j}) + \varepsilon_{i,j} \quad \text{where} \quad \varepsilon_{i,j} \sim \mathcal{N}(0, \sigma^2).$$

The choice of the isometries $\phi_0^\ell$ and the diffeomorphisms $\phi_i^\ell$ will induce a large panel of piecewise-geodesic models. For example, if $m = 1$, $\phi_0 = Id$ and if $\phi_i^1$ denotes the application that maps the curve $\gamma_0$ onto its parallel curve for a given non-zero tangent vector $w_i$, we feature the model proposed by Schiratti et al. (2015). In the following paragraph we propose another specific model which can be used for chemotherapy monitoring for instance (see section 4.2).

## 2.2 Piecewise-logistic curve model

We focus in the following on the case of piecewise-logistic model, which presents a real interest regarding to our target application (cf section 4.2). We assume that $m = 2$ and $d = 1$ and we set $M_0 = ]0, 1[$ equipped with the logistic metric. Given three real numbers $\gamma_0^{\text{init}}$, $\gamma_0^{\text{escap}}$ and $\gamma_0^{\text{fin}}$ we set down $\phi_0^1 \colon x \mapsto \left(\gamma_0^{\text{init}} - \gamma_0^{\text{escap}}\right) x + \gamma_0^{\text{escap}}$ and $\phi_0^2 \colon x \mapsto \left(\gamma_0^{\text{fin}} - \gamma_0^{\text{escap}}\right) x + \gamma_0^{\text{escap}}$. Thus, we can map $M_0$ onto the intervals $]\gamma_0^{\text{escap}}, \gamma_0^{\text{init}}[$ and $]\gamma_0^{\text{escap}}, \gamma_0^{\text{fin}}[$ respectively: if $\bar\gamma_0$ refers to the sigmoid function, $\phi_0^1 \circ \bar\gamma_0$ will be a logistic curve, growing from $\gamma_0^{\text{escap}}$ to $\gamma_0^{\text{init}}$.

In this way, there is essentially a single breaking-up time and we will denote it $t_R$ at the population level and $t_R^i$ at the individual one. Moreover, due to our target applications, we force the first logistic to be decreasing and the second one increasing (this condition may be relaxed). Logistics are defined on open intervals, with asymptotic constraints. We want to formulate our constraints on some non-infinite time-points, as explained in the previous paragraph, we set a positive threshold $\nu$ close to zero and demand the logistics $\gamma_0^1$ and $\gamma_0^2$ to be $\nu$-near from their corresponding asymptotes. More precisely, we impose the average trajectory $\gamma_0$ to be of the form $\gamma_0 = \gamma_0^1 \mathbb{1}_{]-\infty, t_R]} + \gamma_0^2 \mathbb{1}_{]t_R, +\infty[}$ where

$$\gamma_0^1 : \mathbb{R} \to ]\gamma_0^{\text{escap}}, \gamma_0^{\text{init}}[ \qquad \gamma_0^2 : \mathbb{R} \to ]\gamma_0^{\text{escap}}, \gamma_0^{\text{fin}}[ \qquad \begin{cases} \gamma_0^{\text{escap}} + 2\nu \leqslant \gamma_0^{\text{init}} \\ \gamma_0^{\text{escap}} + 2\nu \leqslant \gamma_0^{\text{fin}} \end{cases}$$

$$t \mapsto \frac{\gamma_0^{\text{init}} + \gamma_0^{\text{escap}}\, \mathrm{e}^{(at+b)}}{1 + \mathrm{e}^{(at+b)}} \qquad t \mapsto \frac{\gamma_0^{\text{fin}} + \gamma_0^{\text{escap}}\, \mathrm{e}^{-(ct+d)}}{1 + \mathrm{e}^{-(ct+d)}}$$

and $a$, $b$, $c$ and $d$ are some positive numbers given by the following constraints

$$\gamma_0^1(t_0) = \gamma_0^{\text{init}} - \nu\,, \quad \gamma_0^1(t_R) = \gamma_0^2(t_R) = \gamma_0^{\text{escap}} + \nu \quad \text{and} \quad \gamma_0^2(t_1) = \gamma_0^{\text{fin}} - \nu\,.$$

In our context, the initial time of the process is known: it is the beginning of the treatment. So, we assume that the average initial time $t_0$ is equal to zero. Especially $t_0$ is no longer a variable. Moreover, for each individual $i \in [\![1, n]\!]$, the time-warps write $\psi_i^1(t) = \alpha_i^1(t - t_0 - \tau_i^1) + t_0$ and $\psi_i^2(t) = \alpha_i^2(t - t_R - \tau_i^2) + t_R$ where $\tau_i^2 = \tau_i^1 + \left(\frac{1 - \alpha_i^1}{\alpha_i^1}\right)(t_R - t_0)$. From now on, we note $\tau_i$ for $\tau_i^1$.

In the same way as the time-warp, the diffeomorphisms $\phi_i^1$ and $\phi_i^2$ are chosen to allow different amplitudes and rupture values: for each subject $i \in [\![1, n]\!]$, given the two scaling factors $r_i^1$ and $r_i^2$ and the space-shift $\delta_i$, we define $\phi_i^\ell(x) = r_i^\ell (x - \gamma_0(t_R)) + \gamma_0(t_R) + \delta_i$, $\ell \in \{1, 2\}$. Other choices are conceivable but in the context of our target applications, this one is appropriate. Mathematically, any regular and injective function defined on $]\gamma_0^{\text{escap}}, \gamma_0^{\text{init}}[$ (respectively $]\gamma_0^{\text{escap}}, \gamma_0^{\text{fin}}[$) is suited.

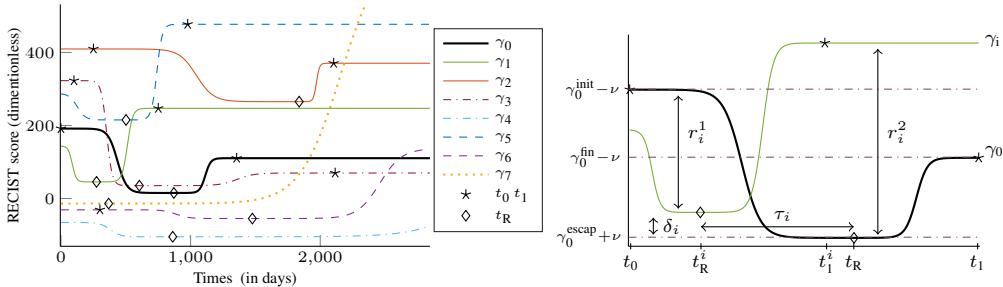

(a) Diversity of individual trajectories.

(b) From average to individual trajectory.

Figure 1: *Model description.* Figure 1a represents a typical average trajectory and several individual ones, for different vectors $P_i$. The rupture times are represented by diamonds and the initial/final times by stars. Figure 1b illustrates the non-standard constraints for $\gamma_0$ and the transition from the average trajectory to an individual one: the trajectory $\gamma_i$ is subject to a temporal and a spacial warp. In other "words", $\gamma_i = \phi_i^1 \circ \gamma_0^1 \circ \psi_i^1 \mathbb{1}_{]-\infty, t_R^i]} + \phi_i^2 \circ \gamma_0^2 \circ \psi_i^2 \mathbb{1}_{]t_R^i, +\infty[}$.

To sum up, each individual trajectory $\gamma_i$ depends on the average curve $\gamma_0$ through fixed effects $z_{\text{pop}} = \left(\gamma_0^{\text{init}}, \gamma_0^{\text{escap}}, \gamma_0^{\text{fin}}, t_R, t_1\right)$ and random effects $z_i = \left(\alpha_i^1, \alpha_i^2, \tau_i, r_i^1, r_i^2, \delta_i\right)$. This leads to a non-linear mixed-effects model. More precisely, for all $(i,j) \in [\![1,n]\!] \times [\![1,k_i]\!]$,

$$y_{i,j} = \left[\, r_i^1 \left(\gamma_i^1(t_{i,j}) - \gamma_0(t_R)\right) + \gamma_0(t_R) + \delta_i \,\right] \mathbb{1}_{]-\infty, t_R^i]}(t_{i,j})$$
$$+ \left[\, r_i^2 \left(\gamma_i^2(t_{i,j}) - \gamma_0(t_R)\right) + \gamma_0(t_R) + \delta_i \,\right] \mathbb{1}_{]t_R^i, +\infty[}(t_{i,j}) + \varepsilon_{i,j}$$

where $\gamma_i^1 = \phi_i^1 \circ \gamma_0^1$, $\gamma_i^2 = \phi_i^2 \circ \gamma_0^2$ and $t_R^i = t_0 + \tau_i + \frac{t_R - t_0}{\alpha_i^1}$. Figure 1 provides an illustration of the model. On each subfigure, the bold black curve represents the average trajectory $\gamma_0$ and the colour curves several individual trajectories.

The acceleration and the scaling parameters have to be positive and equal to one on average while the time and space shifts are of any signs and must be zero on average. For these reasons, we set $\alpha_i^\ell = e^{\xi_i^\ell}$ and $r_i^\ell = e^{\rho_i^\ell}$ for $\ell \in \{1, 2\}$ leading to $P_i = {}^t\!\left(\, \xi_i^1\ \xi_i^2\ \tau_i\ \rho_i^1\ \rho_i^2\ \delta_i \,\right)$. We assume that $P_i \sim \mathcal{N}(0, \Sigma)$ where $\Sigma \in \mathscr{S}_p\mathbb{R}$, $p = 6$. This assumption is important in view of the applications. Usually, the random effects are studied independently. Here, we are interested in correlations between the two phases of patient's response to treatment (see section 4.2).

# 3 Parameters estimation with the MCMC-SAEM algorithm

In this section, we explain how to use a stochastic version of the EM algorithm to produce *maximum a posteriori* estimates of the parameters.

## 3.1 Statistical analysis of the piecewise-logistic curves model

We want to estimate $(z_{\text{pop}}, \Sigma, \sigma)$. The theoretical convergence of the EM algorithm, and *a fortiori* of the SAEM algorithm (Delyon et al., 1999), is proved only if the model belongs to the curved exponential family. Moreover, for numerical performances this framework is important. Without further hypothesis, the piecewise-logistic model does not satisfy this constraint. We proceed as in Kuhn and Lavielle (2005): we assume that $z_{\text{pop}}$ is the realization of independent Gaussian random variables with fixed small variances and estimate the means of those variables. So, the parameters we want to estimate are from now on $\theta = \left(\overline{\gamma_0^{\text{init}}}, \overline{\gamma_0^{\text{escap}}}, \overline{\gamma_0^{\text{fin}}}, \overline{t_R}, \overline{t_1}, \Sigma, \sigma\right)$.

The fixed and random effects $z = (z_{\text{pop}}, z_i)_{1 \leqslant i \leqslant n}$ are considered as latent variables. Our model write in a hierarchical way as

$$\begin{cases} y \,|\, z, \theta \sim \bigotimes_{i=1}^{n} \bigotimes_{j=1}^{k_i} \mathcal{N}\left(\gamma_i(t_{i,j}), \sigma^2\right) \\[2ex] z \,|\, \theta \sim \mathcal{N}(\overline{\gamma_0^{\text{init}}}, \sigma_{\text{init}}^2) \otimes \mathcal{N}(\overline{\gamma_0^{\text{escap}}}, \sigma_{\text{escap}}^2) \otimes \mathcal{N}(\overline{\gamma_0^{\text{fin}}}, \sigma_{\text{fin}}^2) \otimes \mathcal{N}(\overline{t_R}, \sigma_R^2) \otimes \mathcal{N}(\overline{t_1}, \sigma_1^2) \bigotimes_{i=1}^{n} \mathcal{N}(0, \Sigma) \end{cases}$$

where $\sigma_{\text{init}}, \sigma_{\text{escap}}, \sigma_{\text{fin}}, \sigma_R$ and $\sigma_1$ are hyperparameters of the model. The product measures $\otimes$ mean that the corresponding entries are considered to be independent in our model. Of course, it is not the case for the observations which are obtained by repeating measurements for several individuals but this assumption leads us to a more computationally tractable algorithm.

In this context, the EM algorithm is very efficient to compute the *maximum* likelihood estimate of $\theta$. Due to the non-linearity of our model, a stochastic version of the EM algorithm is adopted, namely the Stochastic Approximation Expectation-Maximization (SAEM) algorithm. As the conditional distribution $q(z|y, \theta)$ is unknown, the Expectation step is replaced using a Monte-Carlo Markov Chain (MCMC) sampling algorithm, leading to consider the MCMC-SAEM algorithm introduced in Kuhn and Lavielle (2005) and Allassonnière et al. (2010). It alternates between a simulation step, a stochastic approximation step and a maximization step until convergence. The simulation step is achieved using a symmetric random walk Hasting-Metropolis within Gibbs sampler (Robert and Casella, 1999). See the supplementary material for details about algorithmics.

To ensure the existence of the *maximum a posteriori* (theorem 1), we use a "partial" Bayesian formalism, *i.e.* we assume the following prior

$$(\Sigma, \sigma) \sim \mathcal{W}^{-1}(V, m_\Sigma) \otimes \mathcal{W}^{-1}(v, m_\sigma) \quad \text{where} \quad V \in \mathscr{S}_p\mathbb{R}, \ v, m_\Sigma, m_\sigma \in \mathbb{R}$$

and $\mathcal{W}^{-1}(V, m_\Sigma)$ denotes the inverse Wishart distribution with scale matrix $V$ and degrees of freedom $m_\Sigma$. In order for the inverse Wishart to be non-degenerate, the degrees $m_\Sigma$ and $m_\sigma$ must satisfy $m_\Sigma > 2p$ and $m_\sigma > 2$. In practice, we yet use degenerate priors but with correct posteriors .To be consistent with the one-dimension inverse Wishart distribution, we define the density function of distribution of higher dimension as

$$f_{\mathcal{W}^{-1}(V, m_\Sigma)}(\Sigma) = \frac{1}{\Gamma_p\left(\frac{m_\Sigma}{2}\right)} \left( \frac{\sqrt{|V|}}{2^{\frac{p}{2}} \sqrt{|\Sigma|}} \exp\left( -\frac{1}{2} \operatorname{tr}\left( V \Sigma^{-1} \right) \right) \right)^{m_\Sigma}$$

where $\Gamma_p$ is the multivariate gamma function. The maximization step is straightforward given the sufficient statistics of our exponential model: we update the parameters by taking a barycenter between the corresponding sufficient statistic and the prior. See the supplementary material for explicit equations.

### 3.2 Existence of the *Maximum a Posteriori*

The next theorem ensures us that the model is well-posed and that the *maximum* we are looking for through the MCMC-SAEM algorithm exists. Let $\Theta$ the space of admissible parameters :

$$\Theta = \left\{ \left( \overline{\gamma_0^{\text{init}}}, \overline{\gamma_0^{\text{escap}}}, \overline{\gamma_0^{\text{fin}}}, \overline{t_R}, \overline{t_1}, \Sigma, \sigma \right) \in \mathbb{R}^5 \times \mathscr{S}_p\mathbb{R} \times \mathbb{R}^+ \ \middle| \ \Sigma \text{ positive-definite} \right\} .$$

**Theorem 1** (Existence of the MAP). *Given the piecewise-logistic model and the choice of probability distributions for the parameters and latent variables of the model, for any dataset* $(t_{i,j}, y_{i,j})_{i \in [\![1,n]\!], \, j \in [\![1,k_i]\!]}$, *there exist* $\widehat{\theta}_{MAP} \in \underset{\theta \in \Theta}{argmax} \, q(\theta | y)$.

A detailed proof is postponed to the supplementary material.

## 4 Experimental results

The piecewise-logistic model has been designed for chemotherapy monitoring. More specifically, we have met radiologists of the Hôpital Européen Georges-Pompidou (HEGP – Georges Pompidou European Hospital) to design our model. In practice, patients suffer from the metastatic kidney cancer and take a drug each day. Regularly, they come to the HEGP to check the tumor evolution. The response to a given treatment has generally two distinct phases: first, tumor's size reduces; then, the tumor grows again. A practical question is to quantify the correlation between both phases and to determine as accurately as possible the individual rupture times $t_R^i$ which are related to an escape of the patient's response to treatment.

### 4.1 Synthetic data

In order to validate our model and numerical scheme, we first run experiments on synthetic data.

We well understood that the covariance matrix $\Sigma$ gives a lot of information on the health status of a patient: pace and amplitude of tumor progression, individual rupture times... Therefore, we have to pay special attention to the estimation of $\Sigma$ in this paragraph.

An important point was to allow a lot of different individual behaviors. In our synthetic example, Figure 1a illustrates this variability. From a single average trajectory ($\gamma_0$ in bold plain line), we can generate individuals who are cured at the end (dot-dashed lines: $\gamma_3$ and $\gamma_4$), some whose response to the treatment is bad (dashed lines: $\gamma_5$ and $\gamma_6$), some who only escape (no positive response to the treatments – dotted lines: $\gamma_7$). Likewise, we can generate "patients" with only positive responses or no response at all. The case of individual $4$ is interesting in practice: the tumor still grows but so slowly that the growth is negligible, at least in the short-run.

Figure 2 illustrates the qualitative performance of the estimation. We are notably able to understand various behaviors and fit subjects which are far from the average path, such as the orange and the green curves. We represent only five individuals but 200 subjects have been used to perform the estimation.

To measure the influence of the sample size on our model/algorithm, we generate synthetic datasets of various size and perform the estimation 50 times for each dataset. Means and standard deviations

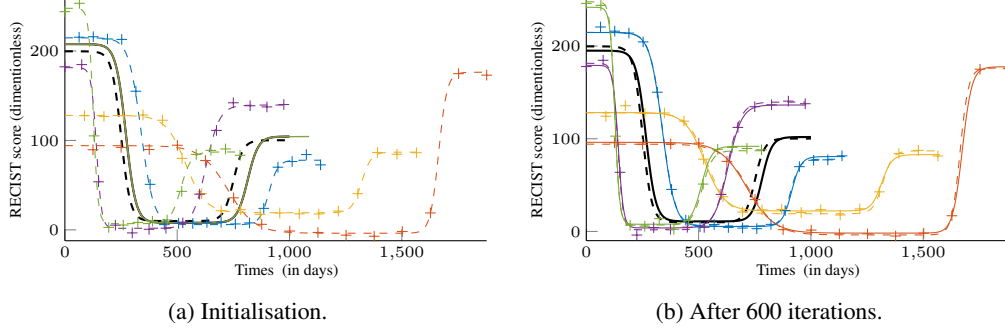

| (a) Initialisation. | (b) After 600 iterations. |

Figure 2: *Initialisation and "results"*. On both figures, the estimated trajectories are in plain lines and the target curves in dashed lines. The (noisy) observations are represented by crosses. The average path is in bold black line, the individuals in color. Figure 2a: Population parameters $\overline{z_{pop}}$ and latent variables $z_{pop}$ are initialized at the empirical mean of the observations; individual trajectories are initialized on the average trajectory ($P = 0$, $\Sigma = 0.1I_p$, $\sigma = 1$). Figure 2b: After 600 iterations, sometime less, the estimated curves fit very well the observations. As the algorithm is stochastic, estimated curves – and effectively the individuals – still wave around the target curves.

Table 1: Mean (standard deviation) of relative error (expressed as a percentage) for the population parameters $\overline{z_{pop}}$ and the residual standard deviation $\sigma$ for 50 runs according to the sample size $n$.

| Sample size $n$ | $\overline{\gamma_0^{\text{init}}}$ | $\overline{\gamma_0^{\text{escap}}}$ | $\overline{\gamma_0^{\text{fin}}}$ | $\overline{t_R}$ | $\overline{t_1}$ | $\sigma$ |
|---|---|---|---|---|---|---|
| 50 | 1.63 (1.46) | 9.45 (5.40) | 6.23 (2.25) | 11.58 (1.64) | 4.41 (0.75) | 25.24 (12.84) |
| 100 | 2.42 (1.50) | 9.07 (5.19) | 7.82 (2.43) | 13.62 (1.31) | 5.27 (0.60) | 10.35 (3.96) |
| 150 | 2.14 (1.17) | 11.40 (5.72) | 5.82 (2.55) | 9.24 (1.63) | 3.42 (0.71) | 2.83 (2.31) |

of the relative errors for the real parameters, namely $\overline{\gamma_0^{\text{init}}}$, $\overline{\gamma_0^{\text{escap}}}$, $\overline{\gamma_0^{\text{fin}}}$, $\overline{t_R}$, $\overline{t_1}$ and $\sigma$, are compiled in Table 1. To compare things which are comparable, we have generated a dataset of size 200 and shortened them to the desired size. Moreover, to put the algorithm on a more realistic situation, the synthetic individual times are non-periodically spaced, individual sizes vary between 12 and 18 and the observed values are noisy ($\sigma = 3$).

We remark that our algorithm is stable and that the bigger the sample size, the better we learn the residual standard deviation $\sigma$. The parameters $\overline{t_R}$ and $\gamma_0^{\text{escap}}$ are quite difficult to learn as they occur on the flat section of the trajectory. However, the error we made is not crippling as the most important for clinicians is the dynamic along both phases. As the algorithm enables to estimate both the mean trajectory and the individual dynamic, it succeeds in estimating the inter-individual variability. This ends in a good estimate of the covariance matrix $\Sigma$ (see figure 4).

## 4.2   Chemotherapy monitoring: RECIST score of treated patients

We now run our estimation algorithm on real data from HEGP.

The RECIST (Response Evaluation Criteria In Solid Tumors) score (Therasse et al., 2000) measures the tumoral growth and is a key indicator of the patient survival. We have performed the estimation over a drove of 176 patients of the HEGP. There is an average of 7 visits per subjects (min: 3, max: 22), with an average duration of 90 days between consecutive visits.

We have run the algorithm several times, with different proposal laws for the sampler (a Symmetric Random Walk Hasting-Metropolis within Gibbs one) and different priors. We present here a run with a low residual standard variation in respect to the amplitude of the trajectories and complexity of the dataset: $\sigma = 14.50$ versus $\max(\gamma_0^{\text{init}}, \gamma_0^{\text{fin}}) - \gamma_0^{\text{escap}} = 452.4$. Figure 3a illustrates the performance of the model on the first eight patients. Although we cannot explain all the paths of progression, the algorithm succeeds in fitting various types of curves: from the yellow curve $\gamma_3$ which is rather flat and only escape to the red $\gamma_7$ which is spiky. From Figure 3b, it seems that the rupture times occur early in the progression in average. Nevertheless , this result is to be considered with some reserve: the rupture time generally occurs on a stable phase of the disease and the estimation may be difficult.

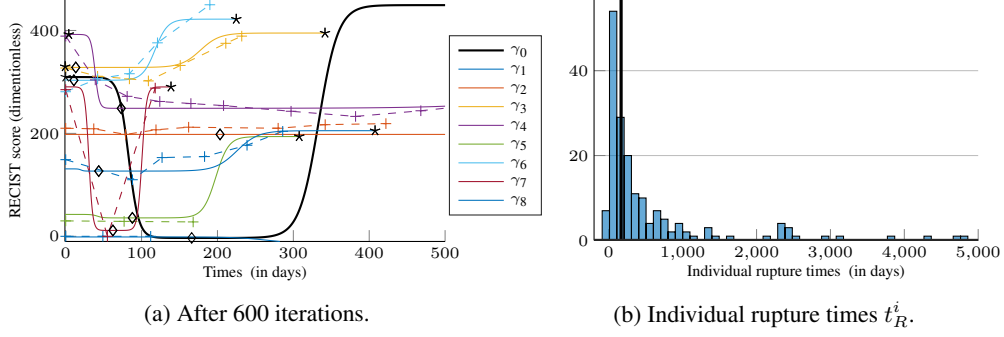

(a) After 600 iterations.
(b) Individual rupture times $t_R^i$.

Figure 3: *RECIST score*. We keep conventions of the previous figures. Figure 3a is the result of a 600 iterations run. We represent here only the first 8 patients among the 176. Figure 3b is the histogram of the rupture times $t_R^i$ for this run. In black bold line, the estimated average rupture time $t_R$ is a good estimate of the average of the individual rupture times although there exists a large range of escape.

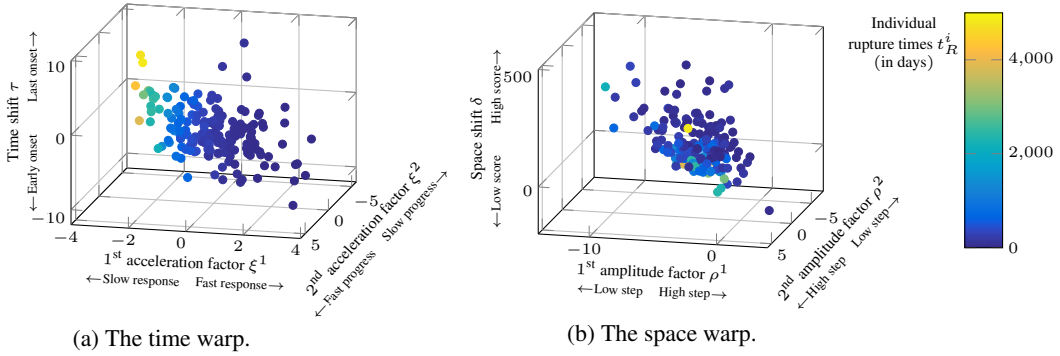

(a) The time warp.
(b) The space warp.

Figure 4: *Individual random effects*. Figure 4a: log-acceleration factors $\xi_i^1$ and $\xi_i^2$ against times shifts $\tau_i$. Figure 4b: log-amplitude factors $\rho_i^1$ and $\rho_i^2$ against space shifts $\delta_i$. In both figure, the color corresponds to the individual rupture time $t_R^i$. These estimations hold for the same run as Figure 3.

In Figure 4, we plot the individual estimates of the random effects (obtained from the last iteration) in comparison to the individual rupture times. Even though the parameters which lead the space warp, *i.e.* $\rho_i^1$, $\rho_i^2$ and $\delta_i$ are correlated, the correlation with the rupture time is not clear. In other words, the volume of the tumors seems to not be relevant to evaluate the escape of a patient. On the contrary, which is logical, the time warp strongly impacts the rupture time.

### 4.3 Discussion and perspective

We propose here a generic spatiotemporal model to analyze longitudinal manifold-valued measurements. Contrary to Schiratti et al. (2015), the average trajectory is not assumed to be geodesic anymore. This allows us to apply our model to more complex situations: in chemotherapy monitoring for example, where the patients are treated and tumors may respond, stabilize or progress during the treatment, with different conducts for each phase. At the age of personalized medicine, to give physicians decision support systems is really important. Therefore learning correlations between both phases is crucial. This has been taken into account here.

For purpose of working with more complicated data, medical images for instance, we have first presented our model in a very generic version. Then we made it explicit for RECIST scores monitoring and performed experiments on data from the HEGP. However, we have studied that dataset as if all patients behave similarly, which is not true in practice. We believe that a possible amelioration of our model is to put it into a mixture model.

Lastly, the SAEM algorithm is really sensitive to initial conditions. This phenomenon is emphasized by the non-independence between the individual variables: actually, the average trajectory $\gamma_0$ is not exactly the trajectory of the average parameters. Fortunately, the more the sample size $n$ increases, the more $\gamma_0$ draws closer to the trajectory of the average parameters.

**Acknowledgments**

Ce travail bénéficie d'un financement public Investissement d'avenir, reference ANR-11-LABX-0056-LMH. This work was supported by a public grant as part of the Investissement d'avenir, project reference ANR-11-LABX-0056-LMH.

Travail réalisé dans le cadre d'un projet financé par la Fondation de la Recherche Médicale, "DBI20131228564". Work performed as a part of a project funded by the Fondation of Medical Research, grant number "DBI20131228564".

## Footnotes

[1] By defining $A^\ell = \phi_0^\ell(\bar{A}^\ell)$ for each $\ell$ we can apply the constraints on $\gamma_0^\ell$ instead of $\bar{\gamma}_0^\ell$.

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
