[Supplementary Material]

# Learning spatiotemporal piecewise-geodesic trajectories from longitudinal manifold-valued data. Supplementary material

**Juliette Chevallier**
CMAP, École polytechnique
juliette.chevallier@polytechnique.edu

**Pr Stéphane Oudard**
Oncology Department
USPC, AP-HP, HEGP

**Stéphanie Allassonnière**
CRC, Université Paris Descartes
stephanie.allassonniere@parisdescartes.fr

## 1 Details about the MCMC-SAEM algorithm

Here, we explicit the MCMC-SAEM algorithm we are use to perform the experiments. We recall that

$$
\begin{aligned}
y_{i,j} \;=\; & \left[\, r_i^1 \left( \gamma_i^1(t_{i,j}) - \gamma_0(t_R) \right) + \gamma_0(t_R) + \delta_i \,\right] \mathbb{1}_{]-\infty, t_R^i]}(t_{i,j}) \\
& + \left[\, r_i^2 \left( \gamma_i^2(t_{i,j}) - \gamma_0(t_R) \right) + \gamma_0(t_R) + \delta_i \,\right] \mathbb{1}_{]t_R^i, +\infty[}(t_{i,j}) \;+\; \varepsilon_{i,j}
\end{aligned}
$$

where

$$
\gamma_0^{\text{init}} \sim \mathcal{N}\left( \overline{\gamma_0^{\text{init}}}, \sigma_{\text{init}}^2 \right) \quad ; \quad \gamma_0^{\text{échap}} \sim \mathcal{N}\left( \overline{\gamma_0^{\text{échap}}}, \sigma_{\text{échap}}^2 \right) \quad ; \quad \gamma_0^{\text{fin}} \sim \mathcal{N}\left( \overline{\gamma_0^{\text{fin}}}, \sigma_{\text{fin}}^2 \right)
$$

$$
t_R \sim \mathcal{N}(\overline{t_R}, \sigma_R^2) \quad ; \quad t_1 \sim \mathcal{N}(\overline{t_1}, \sigma_1^2) \quad ; \quad P_i \overset{i.i.d}{\sim} \mathcal{N}\left(0, \Sigma\right)
$$

and $\theta = \left( \overline{\gamma_0^{\text{init}}}, \overline{\gamma_0^{\text{échap}}}, \overline{\gamma_0^{\text{fin}}}, \overline{t_R}, \overline{t_1}, \Sigma, \sigma \right) \in \Theta$, the space of admissible parameters.

**Prior distribution** : As explain in the article, according to the proof of the existence of the MAP (see bellow), there is no need to put prior on the population parameters. Thus,

$$
q_{prior}(\theta) \;\propto\; \left( \frac{\sqrt{|V|}}{2^{\frac{p}{2}} \sqrt{|\Sigma|}} \exp\left( -\frac{1}{2} \operatorname{tr}\left( V \Sigma^{-1} \right) \right) \right)^{m_\Sigma} \times \left( \frac{v}{\sigma\sqrt{2}} \exp\left( -\frac{v^2}{2\sigma^2} \right) \right)^{m_\sigma} .
$$

**Sufficient statistics** : The complete log-likelihood writes

$$
\begin{aligned}
\log q(,y,z,\theta) \;=\; &-\frac{1}{2}\left[\left(\frac{\gamma_0^{\text{init}}-\overline{\gamma_0^{\text{init}}}}{\sigma_{\text{init}}}\right)^2 + \left(\frac{\gamma_0^{\text{échap}}-\overline{\gamma_0^{\text{échap}}}}{\sigma_{\text{échap}}}\right)^2 + \left(\frac{\gamma_0^{\text{fin}}-\overline{\gamma_0^{\text{fin}}}}{\sigma_{\text{fin}}}\right)^2\right] \\
&-\frac{1}{2}\left[\left(\frac{t_R-\overline{t_R}}{\sigma_R}\right)^2 + \left(\frac{t_1-\overline{t_1}}{\sigma_1}\right)^2\right] \\
&-\frac{m_\Sigma}{2}\sum_{i=1}^{n}\left({}^t P_i\,\Sigma^{-1}\,P_i\right) + \frac{m_\Sigma}{2}\left(\log(|V|)-\log(|\Sigma|)\right) - \frac{1}{2}\operatorname{tr}\left(V\Sigma^{-1}\right) \\
&-\frac{1}{2\sigma^2}\sum_{i=1}^{n}\sum_{j=1}^{k_i}\left(y_{i,j}-\gamma_i(t_{i,j})\right)^2 - \frac{n}{2}\log(|\Sigma|) + m_\sigma\log\left(\frac{v}{\sigma}\right) - \frac{m_\sigma}{2}\left(\frac{v}{\sigma}\right)^2 \\
&+\, constants
\end{aligned}
$$

and thus, we set

$$
\begin{aligned}
S_1(y,z) = \gamma_0^{\text{init}} &\quad;\quad& S_2(y,z) = \gamma_0^{\text{échap}} &\quad;\quad& S_3(y,z) = \gamma_0^{\text{fin}} \\
S_4(y,z) = t_R &\quad;\quad& S_5(y,z) = t_1 &\quad;\quad& S_6(y,z) = \frac{1}{n}\sum_{i=1}^{n}{}^t P_i P_i \;\in\mathcal{M}_p\mathbb{R} \\
\end{aligned}
$$

$$
S_7(y,z) = \frac{1}{k}\sum_{i=1}^{n}\sum_{j=1}^{k_i}\left(y_{i,j}-\gamma_i(t_{i,j})\right)^2.
$$

**Maximisation step** : We simply calculate the partial derivative of the log-likelihood. It comes:

$$
\begin{aligned}
\overline{\gamma_0^{\text{init}}}^{(\texttt{iter}+1)} = S_1(y,z^{(\texttt{iter})}) &\;;\; \overline{\gamma_0^{\text{escap}}}^{(\texttt{iter}+1)} = S_2(y,z^{(\texttt{iter})}) &\;;\; \overline{\gamma_0^{\text{fin}}}^{(\texttt{iter}+1)} = S_3(y,z^{(\texttt{iter})}) \\
\overline{t_R}^{(\texttt{iter}+1)} = S_4(y,z^{(\texttt{iter})}) &\;;\; \overline{t_1}^{(\texttt{iter}+1)} = S_5(y,z^{(\texttt{iter})})
\end{aligned}
$$

and

$$
\Sigma^{(\texttt{iter}+1)} \;=\; \frac{n S_6(y,z^{(\texttt{iter})}) + m_\Sigma V}{n + m_\Sigma} \qquad;\qquad \sigma^{2(\texttt{iter}+1)} \;=\; \frac{k S_7(y,z^{(\texttt{iter})}) + m_\sigma v^2}{k + m_\sigma}.
$$

In particular, the upgraded variances are barycenters between the corresponding sufficient statistics and the priors. Finally, given an adapted sampler (the Symetric Random Walk Hastings-Metropolis within Gibbs Sampler for instance) and the following the sequence $(\varepsilon_{\texttt{iter}})_{\texttt{iter}>0}$

$$
\forall\,\texttt{iter}\geqslant 1,\qquad \varepsilon_{\texttt{iter}} = \begin{cases} 1 & \text{if } \texttt{iter}\geqslant\texttt{Nburnin} \\ (\texttt{iter}-\texttt{Nburnin})^{-0.65} & \text{else} \end{cases}.
$$

our algorithm writes:

---
**Algorithm 1:** Overview of the SAEM for the Piecewise-Logistic model.
---

**Input:** $\theta^* = \left(\overline{\gamma_0^{\text{init}*}}, \overline{\gamma_0^{\text{escap}*}}, \overline{\gamma_0^{\text{fin}*}}, \overline{t_R}^*, \overline{t_1}^*, \Sigma^*, \sigma^*\right), (V, m_\Sigma), (v, m_\sigma), \texttt{maxIter}, \texttt{Nburnin}.$

**Output:** $\theta = \left(\overline{\gamma_0^{\text{init}}}, \overline{\gamma_0^{\text{escap}}}, \overline{\gamma_0^{\text{fin}}}, \overline{t_R}, \overline{t_1}, \Sigma, \sigma\right).$

1   # *Initialization*: $\theta = \left(\overline{\gamma_0^{\text{init}}}, \overline{\gamma_0^{\text{escap}}}, \overline{\gamma_0^{\text{fin}}}, \overline{t_R}, \overline{t_1}, \Sigma, \sigma\right) \leftarrow \theta^*$ ;   $S \leftarrow 0$ ;   $(\varepsilon_{\texttt{iter}})_{\texttt{iter}>0}$ ;

2    $z_{pop} \leftarrow (\overline{\gamma_0^{\text{init}*}}, \overline{\gamma_0^{\text{escap}*}}, \overline{\gamma_0^{\text{fin}*}}, \overline{t_R}^*, \overline{t_1}^*,)$ ;   $(P_i)_i \leftarrow 0$ ;

3 **for** $\texttt{iter} = 1$ **to** $\texttt{maxIter}$ **do**

4    # *Simulation*: $(\gamma_0^{\text{init}}, \gamma_0^{\text{escap}}, \gamma_0^{\text{fin}}, t_R, t_1, (P_i)_i) \leftarrow \texttt{sampler}(\gamma_0^{\text{init}}, \gamma_0^{\text{escap}}, \gamma_0^{\text{fin}}, t_R, t_1, (P_i)_i)$ ;

5    # *Stochastic Approximation*: $S_1 \leftarrow S_1 + \varepsilon_{\texttt{iter}}\left(\gamma_0^{\text{init}} - S_1\right)$ ;
     $S_2 \leftarrow S_2 + \varepsilon_{\texttt{iter}}\left(\gamma_0^{\text{escap}} - S_2\right)$ ;

6     $S_3 \leftarrow S_3 + \varepsilon_{\texttt{iter}}\left(\gamma_0^{\text{fin}} - S_3\right)$ ;    $S_4 \leftarrow S_4 + \varepsilon_{\texttt{iter}}\left(t_R - S_4\right)$ ;
     $S_5 \leftarrow S_5 + \varepsilon_{\texttt{iter}}\left(t_1 - S_5\right)$ ;

7     $S_6 \leftarrow S_6 + \varepsilon_{\texttt{iter}}\left(\frac{1}{n}\sum_i {}^t P_i P_i - S_6\right)$ ;

8     $S_7 \leftarrow S_7 + \varepsilon_{\texttt{iter}}\left(\frac{1}{k}\sum_{i=1}^n \sum_{j=1}^{k_i}\left(y_{i,j} - \gamma_i(t_{i,j})\right)^2 - S_7\right)$ ;

9    # *Maximization*: $\overline{\gamma_0^{\text{init}}} \leftarrow S_1$ ;   $\overline{\gamma_0^{\text{escap}}} \leftarrow S_2$ ;   $\overline{\gamma_0^{\text{fin}}} \leftarrow S_3$ ;   $\overline{t_R} \leftarrow S_4$ ;   $\overline{t_1} \leftarrow S_5$ ;

10    $\Sigma \leftarrow \frac{nS_6 + m_\Sigma V}{n + m_\Sigma}$ ;   $\sigma \leftarrow \sqrt{\frac{kS_7 + m_\sigma v^2}{k + m_\sigma}}$ ;

11 **end**

---

## 2   Proof of the existence of the *Maximum a Posteriori*

**Theorem 1** (Existence of the MAP)**.** *Given the piecewise-logistic model and the choice of probability distributions for the parameters and latent variables of the model, for any dataset* $(t_{i,j}, y_{i,j})_{i \in [\![1,n]\!], j \in [\![1,k_i]\!]}$, *there exists* $\widehat{\theta}_{MAP} \in \underset{\theta \in \Theta}{\text{argmax}}\, q(\theta|y)$.

The demonstration of the theorem uses the following lemma.

**Lemma 1.** *Given the piecewise-logistic model, the choice of probability distribution for the parameters and latent variables of the model, the posterior* $\theta \in \Theta \mapsto q(\theta|y)$ *is continuous on the parameter space* $\Theta$.

*Proof.* Let $\mathcal{Z}$ denote the space of latent variables in the piecewise-logistic model:

$$\mathcal{Z} = \left\{\, (z_{pop}, (z_i)_{1 \leqslant i \leqslant n}) \,|\, z_{pop} \in \mathbb{R}^5,\ \forall i \in [\![1,n]\!],\ z_i \in \mathbb{R}^p \,\right\}$$

Using Bayes rule, for all $\theta \in \Theta$, $q(\theta|y) = \frac{1}{q(y)}\left(\int_{\mathcal{Z}} q(y|z,\theta)\, q(z|\theta)\,\mathrm{d}z\right) q_{prior}(\theta)$. The density function $\theta \mapsto q_{prior}(\theta)$ is trivially continuous on $\Theta$ as a product of continuous functions. Likewise, for all $z \in \mathcal{Z}$, $\theta \mapsto q(y|z,\theta)\, q(z|\theta)$ is continuous. Moreover, for all $z \in \mathcal{Z}$ and $\theta \in \Theta$,

$$q(y|z,\theta) = \frac{1}{(\sigma\sqrt{2\pi})^k}\exp\left(-\frac{1}{\sigma^2}\sum_{i=1}^n \sum_{j=1}^{k_i}(y_{i,j} - \gamma_i(t_{i,j}))^2\right)$$

and so, for all $z \in \mathcal{Z}$ and $\theta \in \Theta$, $q(y|z,\theta)\, q(z|\theta) \leqslant \frac{1}{(\sigma\sqrt{2\pi})^k}\, q(z|\theta)$ which is positive and integrable as a probability distribution function. As a consequence, $z \mapsto q(y|z,\theta)\, q(z|\theta)$ is integrable – and positive – on $\mathcal{Z}$ for all $\theta \in \Theta$ and $\theta \mapsto q(y|\theta)$ is continuous. $\qquad\square$

*Proof of theorem 1.* Given the result of the lemma 1 and considering the Alexandrov one-point compactification $\overline{\Theta} = \Theta \cup \{\infty\}$, it suffices to prove that $\lim_{\theta \to \infty} \log q(\theta|y) = -\infty$. We keep the notation of the previous proof and proceed similarly. In particular, for all $\theta \in \Theta$,

$$\log q(\theta|y) \ \leqslant\ -\log q(y) - k\log(\sqrt{2\pi}) - k\log(\sigma) + \log q_{prior}(\theta)\,.$$

By computing the prior distribution $q_{prior}$, we remark that there exist $C$ which does not depend on the parameter $\theta$ such as

$$\log q(\theta|y) \leqslant C(y) - (k + m_\sigma)\log(\sigma) - \frac{m_\Sigma}{2}\log(|\Sigma|) - \frac{m_\Sigma}{2}\operatorname{tr}\left(V\Sigma^{-1}\right) - \frac{m_\sigma}{2}\left(\frac{v}{\sigma}\right)^2$$

Let $\mu(V)$ denote the smallest eigenvalue of $V$ and $\rho(\Sigma^{-1})$ the largest one of $\Sigma^{-1}$, which is also its operator norm. We know that $\left\langle\, \Sigma \,\middle|\, V \,\right\rangle_F \geqslant \mu(V)\rho(\Sigma^{-1})$ and $\log(|\Sigma^{-1}|) \leqslant p\,\log\left(\,\|\Sigma^{-1}\|\,\right)$ so that

$$-\frac{m_\Sigma}{2}\operatorname{tr}\left(V\Sigma^{-1}\right) + \frac{m_\Sigma}{2}\log(|\Sigma^{-1}|) \leqslant \frac{m_\Sigma}{2}\left[-\mu(V)\,\|\Sigma^{-1}\| + p\,\log\left(\,\|\Sigma^{-1}\|\,\right)\right]$$

and

$$\lim_{\|\Sigma\|+\|\Sigma^{-1}\|\to+\infty}\left\{\, -\frac{m_\Sigma}{2}\operatorname{tr}\left(V\Sigma^{-1}\right) + \frac{m_\Sigma}{2}\log(|\Sigma^{-1}|)\,\right\} = -\infty\,.$$

Likewise,

$$\lim_{\sigma+\sigma^{-1}\to+\infty}\left\{\, -(k + m_\sigma)\log(\sigma) - \frac{m_\sigma}{2}\left(\frac{v}{\sigma}\right)^2\,\right\} = -\infty$$

hence the result. $\qquad\square$

We have detailed the computation in the previous proof in order to emphasize the necessity of prior distribution on the variances $\Sigma$ and $\sigma$ to have the existence of the *maximum a posteriori*.