[Reviews · NeurIPS 2017]

Reviewer 1



This paper describes a mixed modelling approach on longitudinal data embedded in Riemannian manifolds (as opposed to general Euclidean space). The work seems to be an extension of Schiratti et al. (2015), which first introduced this modeling framework. While the time-trajectories in the original work were limited to be geodesics (increasing or decreasing curves in the one-dimensional case), the authors of this manuscript propose to add a breakpoint model that allows for the dynamics to change. They apply this model to the problem of analysing tumour growth data from renal metastatic cancer patients. I will be upfront and say that while I have some knowledge of linear mixed models for longitudinal data in Euclidean spaces, I know next to nothing about working on a manifold. Perhaps for that reason, it was not entirely clear to me what the advantage is. The authors state that "anatomical data are naturally modelled as points on Riemannian manifolds", but why is this? One weakness of this paper in my opinion (and one that does not seem to be resolved in Schiratti et al., 2015 either), is that it makes no attempt to answer this question, either theoretically, or by comparing the model with a classical longitudinal approach. If we take the advantage of the manifold approach on faith, then this paper certainly presents a highly useful extension to the method presented in Schiratti et al. (2015). The added flexibility is very welcome, and allows for modelling a wider variety of trajectories. It does seem that only a single breakpoint was tried in the application to renal cancer data; this seems appropriate given this dataset, but it would have been nice to have an application to a case where more than one breakpoint is advantageous (even if it is in the simulated data). Similarly, the authors point out that the model is general and can deal with trajectories in more than one dimensions, but do not demonstrate this on an applied example. (As a side note, it would be interesting to see this approach applied to drug response data, such as the Sanger Genomics of Drug Sensitivity in Cancer project). Overall, the paper is well-written, although some parts clearly require a background in working on manifolds. The work presented extends Schiratti et al. (2015) in a useful way, making it applicable to a wider variety of datasets. Minor comments: - In the introduction, the second paragraph talks about modelling curves, but it is not immediately obvious what is being modelled (presumably tumour growth). - The paper has a number of typos, here are some that caught my eyes: p.1 l.36 "our model amounts to estimate an average trajectory", p.4 l.142 "asymptotic constrains", p.7 l. 245 "the biggest the sample size", p.7l.257 "a Symetric Random Walk", p.8 l.269 "the escapement of a patient". - Section 2.2., it is stated that n=2, but n is the number of patients; I believe the authors meant m=2. - p.4, l.154 describes a particular choice of shift and scaling, and the authors state that "this [choice] is the more appropriate.", but neglect to explain why. - p.5, l.164, "must be null" - should this be "must be zero"? - On parameter estimation, the authors are no doubt aware that in classical mixed models, a popular estimation technique is maximum likelihood via REML. While my intuition is that either the existence of breakpoints or the restriction to a manifold makes REML impossible, I was wondering if the authors could comment on this. - In the simulation study, the authors state that the standard deviation of the noise is 3, but judging from the observations in the plot compared to the true trajectories, this is actually not a very high noise value. It would be good to study the behaviour of the model under higher noise. - For Figure 2, I think the x axis needs to show the scale of the trajectories, as well as a label for the unit. - For Figure 3, labels for the y axes are missing. - It would have been useful to compare the proposed extension with the original approach from Schiratti et al. (2015), even if only on the simulated data.

Reviewer 2



General comments: This paper describes a mixed effects model of trajectories built with piecewise logistic trajectories and the more general framework within which this model falls. The paper is very interesting and the strengths include a detailed description of a general framework for piecewise-geodesic curve models, a specific model that is fit well for the application selected and a model that includes the covariance of model parameters along with an associated analysis. The weaknesses of this work include that it is a not-too-distant variation of prior work (see Schiratti et al, NIPS 2015), the search for hyperparameters for the prior distributions and sampling method do not seem to be performed on a separate test set, the simultion demonstrated that the parameters that are perhaps most critical to the model's application demonstrate the greatest relative error, and the experiments are not described with adequate detail. This last issue is particularly important as the rupture time is what clinicians would be using to determine treatment choices. In the experiments with real data, a fully Bayesian approach would have been helpful to assess the uncertainty associated with the rupture times. Paritcularly, a probabilistic evaluation of the prospective performance is warranted if that is the setting in which the authors imagine it to be most useful. Lastly, the details of the experiment are lacking. In particular, the RECIST score is a categorical score, but the authors evaluate a numerical score, the time scale is not defined in Figure 3a, and no overall statistics are reported in the evaluation, only figures with a select set of examples, and there was no mention of out-of-sample evaluation. Specific comments: - l132: Consider introducing the aspects of the specific model that are specific to this example model. For example, it should be clear from the beginning that we are not operating in a setting with infinite subdivisions for \gamma^1 and \gamma^m and that certain parameters are bounded on one side (acceleration and scaling parameters). - l81-82: Do you mean to write t_R^m or t_R^{m-1} in this unnumbered equation? If it is correct, please define t_R^m. It is used subsequently and it's meaning is unclear. - l111: Please define the bounds for \tau_i^l because it is important for understanding the time-warp function. - Throughout, the authors use the term constrains and should change to constraints. - l124: What is meant by the (*)? - l134: Do the authors mean m=2? - l148: known, instead of know - l156: please define \gamma_0^{***} - Figure 1: Please specify the meaning of the colors in the caption as well as the text. - l280: "Then we made it explicit" instead of "Then we have explicit it"

Reviewer 3



The paper proposes a mixed effects dynamical model which describes individual trajectories as being generated from transformations of an average behaviour. The work is motivated by modelling of cancer treatment, but the proposed method is more generally applicable. The paper is well-written and -structured, and it considers an interesting application. I have limited knowledge of Riemannian geometry, but the methodology was not hard to follow and appears sound (to the extent that I can judge); that said, I cannot comment on the novelty of the approach. I generally found the presentation to be clear (despite some inconsistencies in the notation). The evaluation is fairly comprehensive and explained well in the text, and the model seems to perform well, especially with regards to capturing a variety of behaviours. It is interesting that the error does not monotonically increase as the sample size increases, but I suppose its variation is not that great anyway. Overall, I believe this is a strong paper that is worthy of acceptance. Most of my comments have to do with minor language or notation issues and are given below. I only have two "major" observations. The first concerns Section 2.2, where the notation changes from the previous section. Specifically, the time warping transformation (previously \psi) is now denoted with \phi, while the space warping (previously \phi) is now strangely \phi(1) (line 154). This is pretty confusing, although presumably an oversight. Secondly, in Figure 1 and its description in Section 4.1, there is no explanation of what the vertical axis represents - presumably the number of cancerous cells, or something similar? Other than that, I think it would be useful to slightly expand the explanation in a couple of places. The first of these is the introduction of the transformations (Section 2.1), where I personally would have welcomed a more intuitive explanation of the time and space warping (although Figure 1b helps with that some). The second place is Section 2.2: here, various decisions are said to be appropriate for the target application (eg lines 140, 155, 166), but without explaining why that is; some details of how these quantities and decisions correspond to the application might be useful. Some of the terminology could also be better explained, especially relating to the treatment, e.g. "escaping", "rupture". Finally, to better place the work in context, the paper could perhaps include a more concrete description of what the proposed model allows (for example, in this particular application) that could not be done with previous approaches (Section 4.3 mentions this very briefly). Detailed comments: ----------------- l21: Actually -> Currently? l22: I don't think "topical" is the appropriate word here; perhaps "contested"? l23: [Mathematics] have -> has l24: [their] efficiency -> its l29: the profile are fitting -> the profiles are fitted? l36: [amounts] to estimate -> to estimating l45: in the aim to grant -> with the aim of granting? l50: temporary [efficient] -> temporarily l54: little [assumptions] -> few l54: to deal with -> (to be) dealt with l55: his -> its l56: i.e -> i.e. Equation between lines 81-82: t_R^m in the subscript in the final term should be t_R^{m-1} (similarly in Equation (*)) l91: I count 2m-2 constraints, not 2m+1, but may be mistaken l93: constrain -> constraint (also repeated a couple of times further down) l106: "we decree the less constrains possible": I am not sure of the meaning here; perhaps "we impose/require the fewest constraints possible"? l118: in the fraction, I think t_{R,i}^{l-1} should be t_R^{l-1} (based on the constraints above) l131: used to chemotherapy monitoring -> either "used for/in chemotherapy monitoring" or "used to [do something] in chemotherapy monitoring" l134: n = 2 -> m = 2 (if I understand right) l139: especially -> effectively / essentially (also further down) l154: \phi_i^l (1), as mentioned earlier in the comments l156: *** -> fin? l160: I think (but may be wrong) that the composition should be the other way around, i.e. \gamma_i^1 = \gamma_0^1 \circ \phi_i^1, if \phi is the time warping here l167: [interested] to -> in l177: independents -> independent l183-185: is it not the case that, given z and \theta, the measurements are (conditionally) independent? l241: the table 1 -> Table 1 Figure 2: there is a thick dark green line with no observations - is that intentional? Table 1: I found the caption a little hard to understand, but I am not sure what to suggest as an improvement l245: the biggest [...], the best -> the bigger [...], the better l257: Symetric -> Symmetric l260: is 14, 50 intended to be 14.50? l260: Figure 3a, -> no comma l264: early on [the progression] -> early in / early on in l264: to consider with quite reserve -> to be considered with some reserve Figure 3: (last line of caption) large scale -> large range? l280: "we have explicit it": I'm not sure what you mean here l284: Last -> Lastly